# Effect of Autoclaving Time on Corrosion Resistance of Sandblasted Ti G4 in Artificial Saliva

**DOI:** 10.3390/ma13184154

**Published:** 2020-09-18

**Authors:** Bożena Łosiewicz, Patrycja Osak, Joanna Maszybrocka, Julian Kubisztal, Sebastian Stach

**Affiliations:** 1Institute of Materials Engineering, Faculty of Science and Technology, University of Silesia in Katowice, 75 Pułku Piechoty 1A, 41-500 Chorzów, Poland; patrycja.osak@us.edu.pl (P.O.); joanna.maszybrocka@us.edu.pl (J.M.); julian.kubisztal@us.edu.pl (J.K.); 2Institute of Biomedical Engineering, Faculty of Science and Technology, University of Silesia in Katowice, Będzińska 39, 41-200 Sosnowiec, Poland; sebastian.stach@us.edu.pl

**Keywords:** autoclaving, corrosion resistance, oxide layer, saliva, titanium

## Abstract

Titanium Grade 4 (Ti G4) is the most commonly used material for dental implants due to its excellent mechanical properties, chemical stability and biocompatibility. A thin, self-passive oxide layer with protective properties to corrosion is formed on its surface. However, the spontaneous TiO_2_ layer is chemically unstable. In this work, the impact of autoclaving time on corrosion resistance of Ti G4 in artificial saliva solution with pH = 7.4 at 37 °C was studied. Ti G4 was sandblasted with white Al_2_O_3_ particles and autoclaved for 30–120 min. SEM, EDS, 2D roughness profiles, confocal laser scanning microscopy, and a Kelvin scanning probe were used for the surface characterization of the Ti G4 under study. In vitro corrosion resistance tests were conducted using open circuit potential, polarization curves, and electrochemical impedance spectroscopy measurements. It was found that Sa parameter, electron work function, and thickness of the oxide layers, determined based on impedance measurements, increased after autoclaving. The capacitive behavior and high corrosion resistance of tested materials were revealed. The improvement in the corrosion resistance after autoclaving was due to the presence of oxide layers with high chemical stability. The optimal Ti G4 surface for dentistry can be obtained by sandblasting with Al_2_O_3_ with an average grain size of 53 µm, followed by autoclaving for 90 min.

## 1. Introduction

Titanium belongs to the vital elements. It is characterized by high corrosion resistance, low specific gravity and mechanical properties similar to bone tissue. This metal does not cause toxic reactions and is highly biocompatible with living tissue. There are also no reports of allergic reactions to titanium. Due to these unique properties, titanium and its alloys are widely used in modern implantology, particularly for the production of dental implants [1,2,3,4]. Other popular metals in dentistry are also silver, gold and palladium. However, studies have shown that these noble metals give a positive response in allergic patch tests, which excludes the possibility of their use on dental implants [5,6]. So far, no material with full biological compatibility has been produced that would ensure complete anastomosis and coupling with the living structures of the human body [3].

The precisely defined and designed surface structure of biomaterials is currently the main development strategy for the latest generation of dental implants. Since Brånemark, called the “father of modern dental implantology”, discovered the osseointegrative properties of titanium in 1952 [7], many attempts have been made to design the perfect surface for fast osteoconductivity and osseointegration, enabling immediate functional loading of the implant and long-term anchoring of implants. A particularly important role in achieving the initial stability of the implant-bone tissue connection plays the chemical and phase composition as well as the topography of the implant surface [8,9]. Surface roughness and porosity has been proven to be an essential requirement for long-term implants that are used in dentistry [1]. These factors determine the good mechanical stability of the implant as well as the proper course of the implant osseointegration process with bone tissue [8,9,10,11,12]. Therefore, the surface of titanium dental implants has been modified over the years [2,4,9,11,13].

The oldest surface of titanium dental implants, which appeared on the market in the 70s of the last century, is the machine surface with the least roughness with a *Ra* of only 0.1–0.8 μm [9,14,15,16,17]. The machine surface is obtained by cutting titanium during implant production and exhibits less osseointegration ability compared to implants with a developed surface. It is an isometric, anisotropic surface with present oriented and parallel furrows. Such a smooth surface limits the osseointegration process by extending the implant’s stabilization time and increasing the risk of post-implantation complications. The latest research in the field of implantoprosthetics shows that stabilization of collagen scaffolding, facilitating the accumulation of bone tissue on the implant surface, occurs more easily on the rough surface of the intraosseous implant. The bone adhesion surface to the porous surface of the implant is thus increased. That is why almost all the latest generation dental implants have a porous surface obtained using various technologies specific for a given system and manufacturer. Therefore, titanium implants with a larger surface development are sought after, which will increase the potential for biomechanical contact at the implant–bone border and affect the rate of protein adsorption [18].

Intensive efforts have been made to develop titanium implants with the surface covered with titanium plasma-sprayed coating (TPS) [19,20], the surface with hydroxyapatite (HA) [13,21,22], the sandblasted surface [23,24,25,26], the double acid-etched surface (DE) [23,25,27,28], the sandblasted and etched surface (sandblasted large-grit acid-etched, SLA) [25,29,30,31], the hydrophilic surface (SLActive) [25,30,32,33], or an oxidized (anodized) surface, which was introduced in 2001 and is referred to as TiUnite [28,34,35,36,37]. Recently, it has also been proposed to modify the surface of titanium implants with biologically active substances [38,39]. The latter type of titanium implant surface is still in the experimental phase.

Currently, the sandblasting is one of the most commonly used processes during the final surface treatment of dental implants. This type of surface is rough, irregular, with numerous craters formed as a result of sandblasting and is characterized by isotropy. The surface development of the sandblasted surface is 34% larger compared to the machine surface [26]. During the sandblasting process, the titanium surface is bombarded with an abrasive, usually corundum, which provides the best osseointegration effects. The grain size of Al_2_O_3_ should be a minimum of 25 μm and a maximum of 250 μm. The best results of osseointegration are obtained by using grain sizes of 25–75 μm. After the sandblasting process, small amounts of Al_2_O_3_ may remain on the titanium surface, which may impede healing of bone tissue. Sandblasting with TiO_2_ of 10–125 μm grain size, calcium phosphate powders in the form of hydroxyapatite, or β-tricalcium phosphate (β-TCP) is also used. As the grain size of the abrasive increases, the surface roughness increases. For the sandblasted surface, *Ra* is 1–3 μm.

In the field of implantoprosthetics, there is also an attempt to shorten the time between implantation and patient loading. Therefore, implants are sought that will guarantee faster healing of bone tissue. There is a tendency to introduce a modification of the surface of titanium implants that will ensure rapid osseointegration, thus enabling immediate functional loading of the implant. Initially, this goal was pursued by increasing the surface roughness of the implants [14,15,16,17]. However, too large a surface development of the implant causes the opposite effects. Optimal surfaces appear to be of average roughness as for the sandblasted surface, and with physico-chemical properties that can stimulate bone tissue to grow.

The type of implant surface is an important, but not the only, factor that has a significant impact on the success of the implantation and on the processes that occur during implant osseointegration. A key role in protecting titanium and its alloys against harmful factors of the biological environment is played by a self-passive oxide layer with strong barrier properties in a biological environment, which is also self-healing. However, it should be noted that the self-passive oxide layer on the surface of titanium and its alloys has a discontinuous structure and is chemically unstable, which is of great importance for the protective function that the passive layer is to perform in a biological environment. Due to the extremely fast rate of formation, the self-passive TiO_2_ layers on titanium easily become amorphous, showing high corrosion resistance, which, however, decreases with the crystallization of the oxide layer [39]. This fact suggests that the surface of titanium and its alloys should be subject to additional modification, which by changing the chemical composition, structure and morphology of the surface will improve the corrosion resistance, bioactivity and osseointegration.

In order to obtain a stable oxide layer ensuring high corrosion resistance of titanium implants, various sterilization methods are used, such as steam autoclave sterilization, sterilization using ethylene oxide and UV radiation [40,41,42,43,44,45,46]. However, the most popular method of sterilization of dental implants and medical devices is autoclaving. This method does not change the chemical composition and structure of the material subjected to sterilization compared to other methods. In hot water or steam, heat is sterilized faster and more efficiently than in dry air due to the several times higher thermal conductivity of water compared to air [45]. Sterilization with UV rays affects changes in the TiO_2_ layer nanostructure [42]. Steam autoclave sterilization is the most effective method of heat exchange, it allows the safety of the medical device during its use, in particular in contact with blood.

Corrosion of the passive oxide layer on the surface of titanium and its alloys in body fluids is an electrochemical process. Hence, electrochemical techniques, and especially impedance techniques, are suitable for testing the corrosion resistance of passive layers. In the system of metallic electrode–passive layer–biological environment, a number of phenomena, such as passive dissolution, transpassive dissolution, uniform corrosion, local corrosion, and adsorption, can occur. Electrochemical impedance spectroscopy (EIS) as the only electrochemical method allows quantitative determination of the kinetics of these processes in in situ tests with simultaneous characteristics of the capacitive behavior of the tested systems. Innovative impedance studies presented in our earlier works have shown that EIS can be proposed as a modern tool in in vitro studies of corrosion resistance of titanium and its alloys in body fluids, which provides information on the detailed mechanisms and kinetics of electrochemical corrosion and the properties of passive layers [40,41,47,48].

The most popular solution used for in vitro corrosion resistance tests of titanium dental implants is saline [24,49]. These biomaterials show higher corrosion resistance than surgical steel and cobalt-chromium alloys [2]. There are only few literature reports on the selection of the appropriate type and sterilization time to increase the corrosion resistance of titanium after the sandblasting process [43,44,46]. Since the corrosion resistance of titanium implants largely depends on the thickness of the oxide layer, the main purpose of the research in this paper was to assess for the first time the effect of steam autoclave sterilization time on the in vitro corrosion resistance of sandblasted titanium implants in artificial saliva solution. Classical corrosion resistance test methods, such as the open circuit potential method and polarization curve method, as well as the complementary EIS method, were used in in vitro tests.

## 2. Materials and Methods

The subject of the study was titanium with a purity grade 4 (Ti G4) produced in the form of a rod (Bibus Metals Ltd., Dąbrowa, Poland). Titanium samples in accordance with ISO 5832–2 [50] and ASTM F67 [51] with the chemical composition given in Table 1 and mechanical properties presented in Table 2, were prepared in the form of discs with a diameter of 10 mm and a height of 5 mm. Table 3 shows the physical properties of Ti G4. Disc-shaped Ti G4 samples were grounded with 80 to 5000# grit SiC paper and polished using final OP-S suspension (Struers Inc., Cleveland, OH, USA) to obtain a mirror-like surface. Then, the samples were cleaned for 20 min in an ultrasonic cleaner in acetone and then in ultra-pure water with a resistivity of 18.2 MΩ cm at 25 °C (Milli-Q Advantage A10 Water Purification System, Millipore SAS, Molsheim, France). The cleaning procedure was repeated twice.

Polished and cleaned Ti G4 samples were sandblasted using white corundum of FEPA Grit F220 [52]. Chemical composition of the corundum used is given in Table 4. The hardness of white Al_2_O_3_ on the Mohs scale was 9.20–9.30. The specific density and the bulk density of white corundum was 3.95(50) and 1.52–1.85 g·cm^−3^, respectively. Sandblasting parameters were pressure 0.6 MPa, sandblasting time 15 s, distance of the sandblasting nozzle from the surface of the blasted titanium about 1.5 cm. Sandblasted Ti G4 samples were sonicated for 20 min sequentially in acetone and then in ultra-pure water to remove corundum residues from the surface.

Next, steam autoclave sterilization was carried out in distilled water vapour at 134 °C at a pressure of 2.2 bar for 30, 60, 90, and 120 min using a fully automatic Autoclave Model GR60 DA of Zealway Instrument Inc. (Xiamen, China). Samples were autoclaved in sterilization sleeves containing chemical indicators in the form of a pink rectangle, which changed to brown after the autoclaving process.

The titanium surface morphology before and after autoclaving was examined by scanning electron microscopy (SEM). Observations were carried out using a JEOL JSM-6480 microscope (JEOL Ltd., Tokyo, Japan) at the voltage of 20 kV and the current intensity of 75 mA, additionally equipped with an energy dispersion spectroscopy (EDS) system, which allowed for the determination of the surface chemical composition.

Confocal laser scanning microscopy (CLSM) was used to assess the surface topography of Ti G4. The research used the LEXT OLS4000 confocal laser measuring microscope (Olympus Polska Ltd., Warszawa, Poland).

The Mitutoyo Surftest SJ-500 profilometer (Mitutoyo Polska Ltd., Wrocław, Poland) was used to carry out the observation of the cross-sectional profile of the sliding track.

Ti G4 samples before and after autoclaving were subjected to electrochemical tests in artificial saliva solution (ASS) of pH = 7.4(1) at 37(1) °C using the Autolab/PGSTAT20 computer-controlled electrochemical system (Metrohm Autolab B.V., Utrecht, The Netherlands).

Chemical composition of the ASS is shown in Table 5. To adjust the pH of the ASS, 4% NaOH solution and 1% C_3_H_6_O_3_ solution were used in accordance with ISO 10271 [49]. Before each measurement, the ASS was deaerated with argon (99.999% purity) for 20 min. The three-electrode system was used in electrochemical studies. The working electrode (WE) was a titanium disk with a geometric surface area of 0.785 cm^2^. One side of the WE and its sidewalls were coated with a non-conductive epoxy resin to protect against contact with the ASS. The counter electrode (CE) was a 4 cm^2^ platinum mesh. The reference electrode (RE) was a saturated calomel electrode (SCE) connected to the ASS using Luggin capillary. Open circuit potential, *E*_OC_, was stabilized for 2 h according to ISO 10271 [49].

Then, the polarization curves in the range of potentials close to the *E*_OC_ were recorded at the polarization rate of *v* = 1 mV s^−1^ and subjected to the Savitzky–Golay smoothing algorithm available in the General Purpose Electrochemical System software [53]. Based on the obtained log |*j|* = f(*E*), the Tafel extrapolation method was carried out and the parameters of corrosion resistance as corrosion potential (*E*_cor_), corrosion current density (*j*_cor_), anodic (*b*_a_) and cathodic (*b*_c_) Tafel slopes, polarization resistance (*R*_p_), and corrosion rate (*CR*) at the *E*_cor_, were determined.

The EIS measurements were performed at the *E*_cor_ in the range of the frequencies studied from 10^5^ to 10^−3^ Hz with frequency resolution 0.003%. Ten frequencies per decade were scanned using an excitation signal in the form of a sine wave with an amplitude of 10 mV. The Kramers–Kronig (K–K) relations were used to assess the correctness of the obtained impedance data [54]. The experimental EIS data were analyzed based on the equivalent electrical circuit concept. Simulation of the equivalent circuit responses and fitting the circuit parameters to the measured data was realized using the EQUIVCRT program [54]. The complex non-linear least squares (CNLS) method with modulus weighting was used. The statistical F test was used to check the importance of the parameters of the selected circuit.

The local contact potential difference (*V*_CPD_) measurements were carried out in air by the scanning Kelvin probe (SKP) method. The PAR Model 370 Scanning Electrochemical Workstation (Princeton Applied Research, Oak Ridge, TN, USA) was used. The distance between the tungsten microprobe-tip and the sample surface was ca. 100 μm. An area of 1000 × 1000 μm^2^ was scanned. The tip-sample arrangement was assumed to be a capacitor, and the *V*_CPD_ was determined from the difference of work function (*WF*) for sample and tip [55]:(1)$$V_{\mathrm{CPD}}=\frac{WF_{\mathrm{sample}}-WF_{\mathrm{tip}}}{e}$$
where *e* denotes the elementary charge.

## 3. Results and Discussion

### 3.1. SEM/EDS Observations

The material in the initial state was the Ti G4 dental metal with machine surface subsequently sandblasted with white Al_2_O_3_ particles with a grain size of 53–75 µm. The average micron size of the grain was 53 µm. The SEM image of Al_2_O_3_ particles used for sandblasting shows the sharp-edged shape of the grains (Figure 1a). This is due to the fact that the grains of very fine abrasive material always break according to specific splitting planes as a result of grinding. Therefore, they have shapes similar to prisms with a triangular or quadrangular base, pyramids with the same base, octagons, and cubes. The use of sharp-edged grains reduces abrasive consumption during sandblasting.

EDS microanalysis allowed to identify the chemical elements contained in the tested white corundum in the form of powder. The EDS spectrum depending on the number of counts as a function of radiation energy for Al_2_O_3_ is shown in Figure 1b. Qualitative EDS analysis confirmed the presence of elements with the atomic number Z of 8 and 13, i.e., oxygen and aluminum, respectively. No other elements that could originate from impurities in the abrasive used for sandblasting of the Ti G4 were revealed. The obtained results confirm that the corundum used in the sandblasting process was characterized by high purity.

The morphology of the Ti G4 machine surface is poorly developed as shown in Figure 2a. On a flat surface, oriented and parallel grooves ≤10 μm can be observed. These scarce morphological motifs were created during the titanium cutting process in which the cutting tool created shallow grooves and left some melted metal. The machine surface is isometric, anisotropic and the least rough of the titanium implant surfaces used in implantology with the average thickness of the self-passive TiO_2_ layer equal to 17 nm [9,14,15,16,17]. Such a little advanced surface is characterized by weaker osseointegration in comparison with implants with more developed surfaces.

Analysis of the local chemical composition of the Ti G4 machine surface showed the presence of peaks originated from elements with the atomic number Z of 8 and 22, i.e., oxygen and titanium, respectively (Figure 2b). Detected peak of O is of very low intensity due to the fact that oxygen is associated with the presence of the self-passive oxide layer on the Ti G4 substrate.

In the next stage of research, sandblasting was carried out, which was aimed at cleaning and leveling the machine surface of Ti G4 and obtaining the appropriate roughness using white corundum under compressed air pressure. Sandblasting as a technological process has been patented by Tilghman in 1870 [56]. The SEM image of the Ti G4 machine surface after sandblasting is shown in Figure 3a. The Ti G4 topography became irregular and rough as a result of the bombardment of the titanium surface with Al_2_O_3_ grains. As a result of plastic deformation, dents are observed in some places on the titanium surface after sandblasting, in others, bulges increase. Microscopic observation of Ti G4 surface shows the presence of small craters and microcracks initiated on the surface during the sandblasting process. This phenomenon is the result of the compressive residual stress after sandblasting and surface hardening [24]. Despite a rougher surface, no macro-scale cracks form on the surface because, after sandblasting, a fine-grained structure is created that limits crack initiation. The sandblasted surface is isotropic without the dominant structure direction. The development of the Ti G4 sandblasted surface compared to the machine surface provides better osseointegration effects [57].

EDS spectrum of the Ti G4 machine surface after sandblasting indicated the presence of Ti and O only (Figure 3b). Very low intensity of the peak originated from O is related to low oxygen content in the self-passive TiO_2_ layer on the sandblasted titanium with the average thickness of 2−5 nm [57]. No peaks originating from Al were detected that would indicate that Al_2_O_3_ particles stuck into the titanium surface during the sandblasting process. Such an unfavorable phenomenon was previously observed in the literature when corundum 90 µm was used for sandblasting titanium dental implants [58]. Alumina particles were identified by energy dispersive X-ray spectroscopy on the sandblasted surface of titanium and attributed to sandblasting residues that could not be removed even after acid etching. Remaining even small amounts of Al_2_O_3_ on the sandblasted surface of titanium can impede bone healing. The result obtained in this study suggests that the proposed conditions of sandblasting ensure the optimum amount of compressive residual stress induced by sandblasting, which is controlled by air pressure and the size and shape of the abrasive particles. The stresses generated harden the titanium and impede the stick of Al_2_O_3_ particles into the surface despite their sharp shapes, as shown in Figure 1a. The surface topography obtained in this way can positively affect cell proliferation, as well as increase the mechanical stability of Ti G4 implants after the implantation process [57].

### 3.2. Surface Roughness

The surface quality of the biomaterial significantly affects the functional properties of dental implants, which are expressed, among others through corrosion resistance. The geometrical structure of the surface (GSS) is one of the most important factors determining the quality of a given surface. It defines a set of all overlapping inequalities resulting from machining processes and material consumption. These inequalities have different shapes, dimensions and location. From a metrological point of view, one of the most important components of the surface layer is roughness which is a predetermined and predictable parameter. The selection of appropriate surface treatment parameters results in the expected roughness. Roughness is of paramount importance in dental implants, as it directly affects their durability. Roughness parameters refer to specific profile features. Full surface characteristics therefore require a set of parameters describing the height of the inequalities, their shape and the spacing between them. These parameters can be distinguished over thirty, while in everyday practice only a few are used.

Detailed characteristics of the GSS for the machine and sandblasted Ti G4 was carried out based on measurements of surface microgeometry in the two-dimensional (2D) system. In order to determine the roughness parameters, the measured profiles were leveled. Symmetrical surface profiles were obtained. Basic surface texture parameters were determined according to ISO 4287 [59] and the results obtained are presented in Table 6. Determined amplitude parameters were separated into two groups depending on the type of profile from which they were calculated. *P* parameters were calculated on the primary profile (raw profile after measurement). *R* parameters were calculated on the roughness profile. Definitions for *P* and *R* parameters are given in ISO 4287 [59]. Expanded surface texture parameters were determined according to ISO 13565–2 [60]. The discussion of the results was limited to profile height parameters with the use of which it is possible to analyze the topography and development of the surface of the tested materials.

The 2D roughness profile for the Ti G4 with machine surface is shown in Figure 4a. The *Psk* parameter meaning skewness of the primary profile is −0.07 which points to a symmetrical profile (Table 6). The arithmetic mean deviation of the roughness profile *Ra* of the machine surface is 0.12 µm, which indicates a smooth surface of the material (Table 6). As shown in Figure 4a, some sites on the surface were as deep as almost 1 µm from the mean line, which was consistent with SEM observations (Figure 2a). The *Ra* parameter is privileged and is most often used. In practice, it is the distance between two parallel lines limiting the surface area of the profile areas filled with material above the average line and the profile areas without material below the average line. It is the height of the space around the average line formed after “leveling” hills and valleys. The *Ra* parameter in μm better reflects the size of the roughness on larger surfaces, as it eliminates the influence of single, irregular hills or pits. The *Rz* parameter, which for the Ti G4 with machine surface was 1.23 µm, is also preferred (Table 6). This is the average of the difference of the five largest heights and the lowest valleys along the length of the elementary segment. The *Rz* parameter is also sometimes referred to as “roughness height by 10 points” and reflects very accurately the roughness tested, which is often a drawback, because the overall roughness image can be distorted by any accidental excessive hill or valley, significantly affecting the measurement result. It seems that *Ra* parameter is better for holistic roughness recognition, while *Rz* parameter is more appropriate for local field assessment.

The graphical interpretation of the research was presented in the form of a percentage distribution of the height ordinate on a unit surface (Figure 4b). The machine surface of Ti G4 is characterized by a symmetrical distribution of the height ordinate. Values of the height ordinate are in the range of 0.448 to 1.7 µm. The highest percentage of about 21% is observed for the height from 0.895 to 0.985 µm. The percentage of the maximum value of the height ordinate on the machine surface is only about 0.1%. The results obtained are in great agreement with the results of SEM observations in Figure 2a, which confirmed the small differentiation of the Ti G4 machine surface topography.

The surface bearing area ratio curve shown in Figure 4c was the basis for the determination of expanded surface texture parameters according to ISO 13565–2 [60] which defines a set of *Rk* parameters that can be calculated from a graphical construction on the Abbot-Firestone curve of a S-shape. Height characterization using the linear material ratio curve included the determination of the following parameters: *Rk*, core roughness depth, *Rpk*, reduced peak height, *Rvk*, reduced valley depth, *Mr1*, material portion 1, *Mr2*, material portion 2, *A1*, area between the upper intersection line and the material ratio curve, and *A2*, the area between the material ratio curve and the bottom intersection line. In Figure 4c, the surface division into three parts is visible: the area of the hills with *Rpk* of 0.161 μm, the area of the profile roughness core for which *Rk* = 0.37 μm, and the area of the profile valleys which is characterized by *Rvk* = 0.206 μm. The low value of the *Rpk* parameter indicates a smooth machine surface of Ti G4. The results obtained indicate that in order to use the Ti G4 surface for dental implants, it is necessary to increase the surface roughness.

The 2D roughness profile for the Ti G4 with sandblasted surface is presented in Figure 5a. After the sandblasting process, a symmetrical profile was obtained, as evidenced by the skewness parameter *Psk*, which is −0.08 (Table 6). As a result of such a surface treatment of the Ti G4, an increase of the arithmetic mean deviation of the roughness profile by over 13 times compared to the *Ra* for machine surface is observed (Table 6). The maximum height of the roughness profile for the sandblasted Ti G4 also increases by over nine times compared to *Rz* obtained for the machine surface (Table 6). Analysis of the roughness profile indicates an increase in the topography development for the surface after sandblasting, for which numerous sites with a depth of almost 10 µm from the mean line are observed in accordance with the SEM observations (Figure 3a).

After the sandblasting process, a surface with symmetrical distribution of the height ordinate was obtained (Figure 5b). The height ordinate values changes from 0 to 14.9 μm. The highest percentage of about 11% is seen for the height ordinate from 5.98 to 8.22 μm. The percentage of the maximum value of the height ordinate on the sandblasted surface is only about 0.1%. These results confirm the SEM observations in Figure 3a, where more advanced topography of Ti G4 sandblasted surface is observed with numerous, evenly distributed, hills and valleys of varying heights.

Figure 5c presents the surface bearing area ratio curve for the Ti G4 after sandblasting. Height characterization using the linear material ratio curve was carried out based on the average height of protruding peaks above roughness core profile of *Rpk* = 1.33 μm, depth of the roughness core profile of *Rk* = 4.55 μm, and average depth of valleys projecting through roughness core profile of *Rvk* = 1.78 μm. The *Rk* parameter increased by more than 12 times and both *Rpk* and *Rvk* parameters more than eight times compared to the smooth machine surface of Ti G4. The obtained results confirm the increase in surface roughness of Ti G4 as a result of sandblasting which is desirable for metallic biomaterials. According to literature data, the optimal *Ra* value between 1 and 2 μm is required [17].

### 3.3. Surface Topography

Surface topography plays a key role in determination of the functional performance of dental implants. However, surface topography characteristics based only on the registration of the 2D roughness profile, described by one or even several height parameters, is insufficient [61]. Complementary three-dimensional (3D) methods for the local measurement and understanding of surface topography are required. Recently, confocal laser scanning microscopy is used extensively in the field of biomaterials, especially for the characterization of surface irregularities and tests of surface topography in hypsometric colors that allow to read the location of hills and valleys in microareas. The depth of focus in the CLSM technique is controlled and highly limited [62].

Figure 6 shows surface topography maps in microareas for the sandblasted Ti G4 before and after sterilization from 30 to 120 min obtained using the CLSM. This optical imaging technique allowed to increase optical resolution and contrast of the micrographs through application of a spatial pinhole in order to block out-of-focus light in image formation. In the optical sectioning process, many 2D images were captured at various depths in the sample tested, which enabled the reconstruction of 3D structures. The surfaces of the Ti G4 subjected to sandblasting and autoclaving (Figure 6b–e) have a microstructure with predominance of deeper valleys compared to the less developed surface before autoclaving (Figure 6a). The results obtained could suggest a greater susceptibility to corrosion of more rough surfaces. However, it should be taken into account that such a clear relationship between 2D/3D surface roughness parameters and a measurable index of functional surface characteristics, such as corrosion resistance, is typical of metals and alloys that are not in a passive state. The research challenge is to correlate the GSS parameters with the functional features of surfaces in the passive state like in case of the tested Ti G4, so that on the basis of these first measurements it is possible to predict the corrosion performance of the manufactured items.

The height parameters of surface roughness have the strongest impact on the intensity of corrosion due to increasing the real contact surface of the corrosive element. Therefore, based on the obtained surface topography maps, the *Sa* parameter value was calculated in accordance with ISO 25178–2 [63]. The *Sa* belonging to the category of height parameters denotes the arithmetic mean of the absolute value of the height from the mean plane of the surface. The determined values of the *Sa* parameter together with standard deviations obtained for the Ti G4 before and after autoclaving are presented in Table 7.

The lowest value of *Sa* = 1.34 µm was obtained for the Ti G4 after sandblasting (Table 7). This 3D parameter has the same physical meaning as *Ra* = 1.65 µm determined during the 2D profilometric examination (Table 6). Therefore, the results obtained on the basis of the roughness profile with sampling length of 4 mm (Figure 5a) and the 130 × 130 µm^2^ surface topography map (Figure 6a) are comparable. This means that the surface topography of the analyzed sample is relatively homogeneous. The *Sa* parameter represents an overall measure of the texture comprising the surface. As a result of steam autoclave sterilization applied this height parameter value slightly increased due to formation of thin oxide layers on the surface of the sandblasted sample (Table 7). The optimal autoclaving time after which the largest development of the surface for the sandblasted Ti G4 was 90 min.

### 3.4. Open Circuit Potential Measurements

The open circuit potential was used as a parameter determining the initial in vitro corrosion resistance of Ti G4 in a biological environment. The OCP method aimed to assess the protective properties of passive layers that arose as a result of sterilization in a steam autoclave on the sandblasted surface of the titanium electrode. Figure 7 illustrates the dependence of the open circuit potential for the sandblasted Ti G4 electrode before and after autoclaving on the immersion time (*t*) in ASS at 37 °C.

The *E*_OC_ changed rapidly during the first 1500 s after immersing all electrodes in the ASS. The ionic-electron equilibrium on the electrode *|* ASS interface was reached after 120 min, and then the stable value of *E*_OC_ was determined which was treated in further considerations as an approximate value of the *E*_cor_. The lowest corrosion resistance was demonstrated by the Ti electrode after sandblasting for which the average value of *E*_OC_ was −0.325(9) V. The most dynamic changes in the *E*_OC_ are observed for this electrode. This fact can be related to the presence of a self-passive oxide layer, which exhibits a discontinuous structure and is chemically unstable, and its barrier properties are insufficient. Initially, the *E*_OC_ significantly decreased over time due to very slow dissolution of the self-passive oxide layer or conversion of hydrated oxides into anhydrous oxides [37]. The *E*_OC_ then increased, suggesting the formation of an oxide layer on the electrode surface in the ASS or the sealing of an existing self-passive oxide layer on the Ti G4 surface. The average *E*_OC_ for all autoclaved titanium electrodes was more positive compared to the *E*_OC_ for the sandblasted Ti G4 before steam autoclave sterilization. Such a character of the *E*_OC_ changes suggests a decrement in the thermodynamic tendency to the corrosion of Ti G4. The average *E*_OC_ value changed with increasing autoclaving time from −0.193(5) V for Ti G4 electrode after 30 min of autoclaving to −0.060(2) V for Ti G4 electrode after 120 min of autoclaving, respectively. The highest average *E*_OC_ of −0.051(2) mV was obtained for the Ti G4 electrode after 90 min of autoclaving, which means that the obtained ultra-thin passive oxide layer revealed the strongest barrier properties. The obtained results show that the application of the autoclaving process has significantly improved the corrosion resistance of sandblasted Ti G4. The measurement conditions of the *E*_OC_ for the Ti G4 electrode before and after steam autoclave sterilization in the ASS at 37 °C are very similar to the in vivo conditions in the human body.

### 3.5. Polarization Curves Near the Open Circuit Potential

The polarization curves for the sandblasted Ti G4 electrodes before and after autoclaving in the ASS at 37 °C are shown in Figure 8. The electrochemical response of the tested electrodes was investigated near their open circuit potential. Within this narrow potential window of ±50 mV versus the *E*_OC_, the relationship between the potential and the logarithm of the current density module is linear. The obtained polarization curves closely mimic steady-state conditions due to a slow scan rate used. This method ensured the immediate measurement without the need to conduct long-term experiments in order to obtain a significant loss of the electrode mass. Corrosion resistance parameters determined based on the Tafel extrapolation of the polarization curves presented in Figure 8 are listed in Table 8.

The lowest average *E*_cor_ value of −0.337(5) V is observed for the sandblasted Ti G4 electrode before steam autoclave sterilization. A slightly higher value of *E*_cor_ was observed for sandblasted 35A commercially pure titanium in a 3.5% NaCl solution at room temperature [24]. However, the authors used SiO_2_ particles of 200–300 µm in diameter and recorded polarization curves with a scanning rate of 20 mV min^−1^ for the sandblasting process. Figure 8 shows that autoclaving of the sandblasted Ti G4 caused the shift of the average *E*_cor_ value towards the anode potentials. This electrochemical behavior is connected with an increase in corrosion resistance of the autoclaved electrodes due to the presence of the formed oxygen layers with stronger protective barrier properties as compared to the self-passive oxide layer. The highest average value of the *E*_cor_ equal to −0.077(4) V was obtained for the sandblasted Ti G 4 after autoclaving for 90 min (Table 8). Based on the determined *E*_cor_ values, it can be expected that the destructive processes will start fastest in the ASS on the sandblasted Ti G4 with a self-passive oxide layer on the electrode surface. These results correlate with the *Sa* parameter, which indicates the largest changes in the surface topography of the Ti G4 electrode after autoclaving for 90 min, probably resulting from an increase in the thickness of the TiO_2_ layer (Table 7).

The highest average *j*_cor_ value of 288(30) nA cm^−2^ is determined for the sandblasted Ti G4 electrode before autoclaving (Table 8). Significantly lower *j*_cor_ values are observed for the sandblasted Ti G4 electrodes subjected to steam autoclave sterilization. Decreasing the average *j*_cor_ value by almost 3–8 times depending on the autoclaving temperature indicates that the dissolution rate of the oxide layers formed by autoclaving occurs much slower as compared to the self-passive TiO_2_ layer. Although *j*_cor_ is directly proportional to the dissolution rate of the oxide layer, it cannot be used as a kinetic parameter to compare the corrosion resistance of the Ti G4 before and after autoclaving.

The determined *b*_a_ Tafel slopes for all tested electrodes take significantly higher values in comparison with the values of *b*_c_ Tafel slopes (Table 8). This proves that the anode processes are faster than the cathode processes occurring at the corrosion potential, which is a mixed potential and both the reduction and oxidation processes affect its value. The Tafel slope depends on the parameters of the transition, including the construction of the transition complex or the number of exchanged electrons [64]. The Tafel slopes of the cathode and anode process can be expressed by formulas:(2)$$b_{c}=-\frac{2.3RT}{\alpha nF}$$
(3)$$b_{a}=\frac{2.3RT}{(1-\alpha )nF}$$
where R is the gas constant equal to 8.314 J K^−1^ mol^−1^, T is the temperature in K, α is the cathodic transfer coefficient, n is the number of electrons exchanged, and F is the Faraday constant equal to 96,500 C mol^−1^. From the values of the *b*_c_ and *b*_a_ coefficients, one can infer the mechanism of the electrode processes taking place.

The sandblasted Ti G4 before and after autoclaving is characterized by passive behavior which is typical of metal (Me) covered by a metal oxide layer. Both the self-passive TiO_2_ layer and titanium oxide layers formed by autoclaving are oxidized very slowly in the ASS or thickening of the oxide layers occurs. The TiO_2_ layer is the kinetic factor which controls the corrosion rate of Ti G4 in the ASS. The biocompatibility of Ti G4 dental metal is related to the stability of the TiO_2_ layer, which limits both the anodic and cathodic reactions and constitutes a physical barrier for ions transferred to the metal surface and an electronic barrier for electrons. During passive dissolution under anaerobic conditions in an aqueous solution, a charge transfer reaction takes place according to the following general reaction [65]:(4)$$\mathrm{Me}+n\cdot H_{2}O↔{\mathrm{Me}(\mathrm{OH})}_{n}+\frac{n}{2}\cdot H_{2}$$
Water plays the role of the oxidation agent. The products of reaction (4) may be metal oxides, hydroxides, or hydrated oxides and gaseous hydrogen.

Taking into account that the oxidation state IV (Ti^4+^) is thermodynamically most stable form, the anodic process of titanium in aqueous solutions should be formally described by the reaction below [66]:(5)$$\mathrm{Ti}+{4H}_{2}O↔{\mathrm{Ti}(\mathrm{OH})}_{4}+2H_{2}$$
However, Ti^4+^ is a very strong Lewis acid and is not able to exist in aqueous solutions due to deprotonations of OH^−^ ions and H_2_O molecules. As a result, instead of the product in the form of Ti(OH)_4_ according to Reaction (5), more or less hydrated titanium(IV) oxide (TiO_2_·nH_2_O) is formed as a stable end corrosion product.

The Reactions (4) and (5) are coupled with the oxidizing agent reduction which consumes electrons from the oxidation reaction. The Equations (6) and (7) take place in a neutral ASS environment:(6)$$O_{2}+{2H}_{2}O+{4e}^{-}\to {4\mathrm{OH}}^{-}$$
(7)$${2H}_{2}O+{2e}^{-}\to H_{2}+{2\mathrm{OH}}^{-}$$
The major cathodic reaction in the AAS solution can be estimated on the basis of the corrosion potential determined for the Ti G4 electrodes before and after autoclaving (Table 8). From the Nernst equation for the hydrogen evolution reaction, $E_{H^{+}{/H}_{2}}=E_{H^{+}{/H}_{2}}^{0}-0.059\mathrm{pH}$ [64] it was found that $E_{H^{+}{/H}_{2}}$ was equal to −0.681 V vs. SCE at pH 7.4. The values of the *E*_cor_ for all tested electrodes were more positive than the determined $E_{H^{+}{/H}_{2}}$ value (Table 8). This suggests that the hydrogen reduction was not a dominant cathodic reaction. In all cases seen in Table 8, the rate of anodic reaction described by Equation (5) is faster than the reduction reactions illustrated by Equations (6) and (7). This suggests that the autoclaving does not change the nature of the electrochemical processes occurring at the Ti G4 electrode in the ASS.

The average value of *R*_p_ = 6.92 × 10^4^ Ω cm^2^ is the lowest for Ti G4 before autoclaving (Table 8). This results probably from a discontinuous structure and insufficient tightness of the self-passive TiO_2_ layer. The use of steam autoclave sterilization increases the polarization resistance by one order of magnitude indicating an improvement of the corrosion resistance of the autoclaved electrodes.

The calculated values of the *CR* at the *E*_cor_ for the Ti G4 before and after autoclaving are inversely proportional to the polarization resistance (Table 8). Among all the tested electrodes, the highest value of *CR* = 2.51 × 10^−3^ mm yr^−1^ is observed for Ti G4 with the self-passive oxide layer and represents the highest material consumption. The corrosion rate related to the dissolution of the TiO_2_ layers formed by autoclaving are one order of magnitude lower (Equation (5)). The passive layers formed by autoclaving more effectively impede the migration of Ti(IV) ions from the surface of Ti G4 to the ASS, thus minimizing corrosion process through reduction in material consumption.

### 3.6. Electrochemical Impedance Spectroscopy Study

Due to the fact that DC methods cannot be used to measure the charge transfer resistance (*R*_ct_) across the interface of Ti G4|oxide layer|ASS, the study of the kinetics and mechanism of corrosion of the tested system was carried out using the complementary EIS method. This AC method has a great advantage over other electrochemical techniques because it allows to verify the quality of experimental data. The tool used to assess the correctness of the recorded data are the Kramers–Kronig relations. If the recorded data agree with the results obtained from the K–K transformation, it means that they are formally correct and meet the conditions of good measurement, that is, causality, linearity, stability, and finiteness. The K–K relations are based on the relation between the imaginary and real parts of the frequency dispersion, where each imaginary part, *Z*_im_, can be calculated from the real part, *Z*_re_, and vice versa. More details regarding the K–K test can be found elsewhere [54,67].

The validity of raw EIS data for the Ti G4 before and after autoclaving has been confirmed using the K–K test. Relative differences Δ_re,i_ and Δ_im,i_ between the experimental impedance data and the fit according to the K–K test plotted against the log of the frequency did not exceed 10%. For all impedance spectra, there was only a slight distribution of disturbances around the frequency axis, which indicated the data corresponded to K–K relations. An exemplary deviation from the compliance with the K–K relations for the sandblasted Ti G4 electrode is shown in the relative differences plot in Figure 9. The residuals in the entire range of tested frequencies are less than 8%, thus the impedance spectra meet the K–K relation, which proves the correctness of the recorded data. A noise distribution around the frequency axis is not observed in the high and medium frequency range. Only few points with large deviations in the range of low frequencies below 0.01 Hz were present, which could not be described by the K–K transform, therefore they were not included in the detailed analysis of EIS spectra.

The experimental EIS data in the form of Bode diagrams marked with symbols are presented in Figure 10. Symbols in Figure 10a,b are the same. Bode diagrams illustrating the dependence of the log of the |*Z*| as a function of the log of the frequency are characterized by a slope of the impedance module in the range of medium frequencies close to −1 (Figure 10a). An increase in the log |*Z*| value is observed at the lowest measuring frequency for all electrodes subjected to autoclaving, which proves the improvement of corrosion resistance as compared to the sandblasted Ti G4 electrode at the initial state in the following ascending order: electrode before autoclaving > electrode after autoclaving for 30 min > electrode after autoclaving for 60 min > electrode after autoclaving for 120 min > electrode after autoclaving for 90 min. Figure 10b shows experimental Bode diagrams in the form of phase angle (*φ*) as a function of the log of the frequency. A plateau is seen in the medium frequency range. The independence of the phase angle from the frequency indicates the passive state Ti G4 provided by the oxide layers. The widest plateau is visible to the electrode after autoclaving for 90 min, which confirms its highest corrosion resistance in the ASS. For all electrodes tested, the maximum values of *φ* are about 10° less than the ideal value of −90°. One time constant is present in the electrical circuit, which is consistent with the literature data for titanium and its biomedical alloys covered by a thin oxide layer [40,47]. High values of |*Z*|_f→0_ (Figure 10a) and *φ* (Figure 10b) confirm the capacitive behavior of material characterized by high corrosion resistance.

In order to interpret the EIS results with respect to the barrier properties of the oxide layer, an approximation of the experimental AC data was carried out using the equivalent electrical circuit of modified Randle’s cell shown in Figure 10c. This model for corrosion allows for simulating the response of the equivalent electrical circuit and then fitting the parameters of the circuit to the experimental EIS data using the CNLS method [54,67]. Each parameter has an assigned physicochemical meaning characterizing the impedance of Ti G4|TiO_2_|ASS interface. *R*_s_ is related to the solution resistance, *R*_ox_ represents the charge transfer resistance through the TiO_2_|ASS interface, and the CPE_dl_ is the constant phase element (CPE) introduced instead of a capacitor which corresponds to the electrical double layer capacitance (*C*_dl_). This procedure is typically used for ease of fitting for metallic materials coated with oxide layers due to the fact that EIS impedance diagrams deviate from the classical Randles’ equivalent electrical circuit [2,40,47,67]. The impedance of the CPE (${\hat{Z}}_{\mathrm{CPE}}$) is defined by the equation below:(8)$${\hat{Z}}_{\mathrm{CPE}}=\frac{1}{T{\left(j\omega \right)}^{\phi }}$$
where *T* is the capacitance parameter of the CPE expressed in F cm^−2^ s ^ϕ^^−1^, and *ϕ* is a CPE exponent related to the constant phase angle, *α* = 90°(1 − *ϕ*), which is dimensionless and takes values ≤1 [67].

Calculations of the average ${\bar{C}}_{\mathrm{dl}}$ value were made using the following equation [68]:(9)$$T={\bar{C}}_{\mathrm{dl}}^{\phi }{\left(\frac{1}{R_{s}}+\frac{1}{R_{\mathrm{ox}}}\right)}^{1-\phi }$$

The fitted data using CNLS method and the electrical equivalent circuit presented in Figure 10c are marked as continuous lines in Figure 10a,b. A very good quality of fitting to the experimental EIS data was obtained. The values of all parameters resulting from the fitting using the equivalent electrical circuit illustrated in Figure 10c to approximate experimental EIS data obtained for the sandblasted Ti G4 electrode before and after autoclaving at the *E*_cor_ in ASS at 37 °C are summarized in Table 9.

The smallest value of *R*_ox_ is determined for the sandblasted Ti G4 electrode before autoclaving (Table 9). The use of steam autoclave sterilization affects the growth of *R*_ox_ by an order of magnitude. The highest value of *R*_ox_ = 8.80(28) × 10^5^ Ω·cm^2^ was determined for the electrode after 90 min of autoclaving. The nature of changes in the *R*_ox_ parameter is in line with the values of *R*_p_ determined based on the Tafel extrapolation method (Table 8), which confirms the correctness of the obtained results. The physico-chemical meaning of the *R*_ox_ parameter can be related to ongoing corrosion. The charge transfer resistance through the TiO_2_|ASS interface according to Equation (5) takes larger values in case of the oxide layer formed by autoclaving, which show stronger barrier properties to aggressive chloride ions compared to the self-passive TiO_2_ layer. The spontaneously formed TiO_2_ layer is the least thermodynamically stable, possibly due to insufficient compactness, impermeability, and continuity of structure.

For all electrodes under test, the *ϕ*_dl_ parameter shows a significant deviation from 1 (Table 9). It was reported that *ϕ*_dl_ is an empirical parameter which is connected with the presence of physical, chemical or geometrical inhomogeneities [67]. Taking into account the obtained results of the GSS for the Ti G4|TiO_2_|ASS system, the origin of the CPE dispersion can be related mainly to the surface microscopic roughness.

The calculated values of the ${\bar{C}}_{\mathrm{dl}}$ parameter are of the order of 10^−5^ F cm^−2^ for the electrodes after autoclaving and decrease compared to that for the Ti G4 electrode before autoclaving, which is evidence of the decreasing electrochemical activity (Table 9). Similar values of the ${\bar{C}}_{\mathrm{dl}}$ were reported in the literature for passivated titanium and its alloys in a biological environment [40,47,48].

The thickness of the oxide layer (*d*_ox_) formed spontaneously and by autoclaving on the surface of Ti G4 electrode was calculated based on the EIS data using the equation given by Birch and Burleigh [69]:(10)$$d_{\mathrm{ox}}=\varepsilon_{0}\varepsilon 2\pi Sf(Z-Z_{0})$$
where *ε*_0_ represent the dielectric constant for free space equal to 8.854 × 10^−12^ F m^−1^, ε is the dielectric constant for TiO_2_ equal to 110, *S* denotes the area of the electrode surface, *f* is the frequency at which the phase angle reaches its maximum, *Z* stands for impedance at the frequency *f*, and *Z*_0_ is the solution resistance. One can see that the calculated value of the *d*_ox_ increases with the autoclaving time from 1.2 nm for the sandblasted Ti G4 electrode with the self-passive oxide layer to 3.0 nm for the electrode after steam autoclave sterilization for 120 nm (Table 9).

### 3.7. Scanning Kelvin Probe Measurements

Effect of autoclaving time on work function for the sandblasted Ti G4 was determined using the SKP, which is a non-invasive method. The SKP is an analytical tool extremely sensitive to changes in the highest atomic layers caused by deposition, absorption, corrosion, wear and atomic displacement. Therefore, SKP can be treated as a “barometer” of the electrical properties of materials [55]. Surface distribution maps of the *WF* for the sandblasted Ti G4, before and after autoclaving, are pictured in Figure 11.

The maps (*z* variable) in Figure 11 were the basis to obtain the corresponding *WF* distribution histograms shown in Figure 12. Histograms were obtained by dividing the *WF* range into 30 intervals that were equal (Δ*W*) and determination of the *WF* value, which lay in the interval of each interval (*n*_i_). The function ρ(*W*_i_) was approximated by *n*_i_/Σ*_i_n*_i_Δ*W*. It was assumed that the Δ*W* interval was small enough, and the ρ(*z*_i_) was approximated by a continuous function ρ(*z*), for which a Gaussian form was assumed. The Gaussian function described by the equation below was used for approximation of each histogram [70,71]:(11)$$y=\frac{A}{\sigma \sqrt{2\pi }}e^{-\frac{{\left(x-\bar{x}\right)}^{2}}{2\sigma^{2}}}$$
where *A* represents a constant of 1, σ is the standard deviation,$\bar{x}$ denotes the average value of the Gaussian distribution expectation, i.e., a location of the *FW* peak value, and *σ*^2^ is the Gaussian distribution variance, i.e., a measure of the distribution width *WF*. The Gaussian distribution curves of the *WF* for the sandblasted Ti G4 before and after autoclaving are shown in Figure 12. The Gaussian fitting parameters of the *WF* are summarized in Table 10.

In the case of the non-autoclaved Ti G4 sample, the average value of *WF* equals to 3.98 eV (Table 10), which is the lowest amount of energy needed to release electrons from the surface of all tested materials. The determined value is close to the electron work function of 3.84 eV for titanium reported by Kumar and co-workers [72]. On the other hand, the *WF* of 4.33 eV for the polycrystalline titanium determined by photoelectric effect was determined [73]. Due to the fact that the *WF* depends on surface cleanliness, the results obtained often cover a wide range for individual metals. After the autoclaving process, the average *WF* increases, reaching the maximum value of 4.31 eV for the sample autoclaved for 90 min. The increase in *WF* value for autoclaved samples may be related to the removal of organic and inorganic contaminant molecules during steam autoclave sterilization. Low values of *σ*^2^ indicate that the *WF* is close to a certain value and only changes slightly on the surface (Table 10). The obtained SKP data are in very good agreement with the electrochemical results.

## 4. Conclusions

On the basis of the obtained results, it can be concluded that sterilization in a steam autoclave is an effective method of surface modification of the sandblasted Ti G4 dental metal, enhancing its corrosion resistance in the ASS with pH of 7.4 at 37 °C. The use of white Al_2_O_3_ particles for sandblasting process makes it possible to obtain the Ti G4 surface with a very high purity and optimal surface roughness for dental applications. The effect of the autoclaving time on in vitro corrosion resistance of the sandblasted Ti G4 was determined in electrochemical tests using the open circuit potential method, polarization curves, and electrochemical impedance spectroscopy. The improvement in the corrosion resistance was found after autoclaving due to the formation of passive oxide layers with high chemical stability. The thickness of the ultrathin oxide layers on the surface of Ti G4 was evaluated based on the AC impedance measurements in the range from 1.2 nm for the self-passive TiO_2_ layer to 3.0 nm for the passive oxide layer obtained after 120 min of autoclaving. The mechanism and kinetics of corrosion of the sandblasted Ti G4 before and after autoclaving was determined based on the EIS measurements fitted by the equivalent electrical circuit of modified Randle’s cell, where *R*_s_ was related to the solution resistance, *R*_ox_ represented the charge transfer resistance through the TiO_2_|ASS interface, and the CPE_dl_ corresponded to the electrical double layer capacitance of the Ti G4–oxide layer–ASS interface. The capacitive behavior of materials characterized by high corrosion resistance was found. The increase in the electron work function after autoclaving confirmed the electrochemical results.

The optimal Ti G4 surface for dental applications can be obtained by sandblasting with white corundum with an average grain size of 53 µm, followed by autoclaving for 90 min. Such a surface should be characterized by the parameter *Sa* of 3.14 µm, the presence of the oxide layer with a thickness of 2 nm, and the electron work function of 4.31 eV, which will provide strong barrier properties to corrosion in ASS defined by: *E*_cor_ of –0.077 V, *R*_p_ of 2.30 × 10^5^ Ω·cm^2^, *CR* at the *E*_cor_ of 4.93 × 10^−4^ mm yr^−1^, and *R*_ox_ of 8.80 × 10^5^ Ω·cm^2^.

## Figures and Tables

**Figure 1 materials-13-04154-f001:**
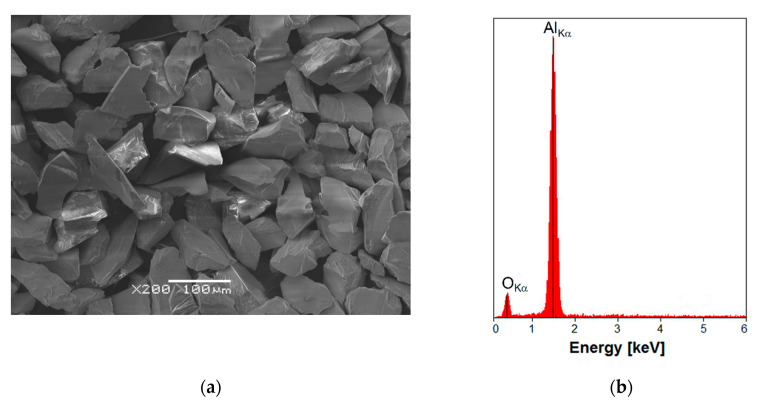
(**a**) SEM image of white Al_2_O_3_ particles used for sandblasting; (**b**) EDS spectrum of Al_2_O_3_.

**Figure 2 materials-13-04154-f002:**
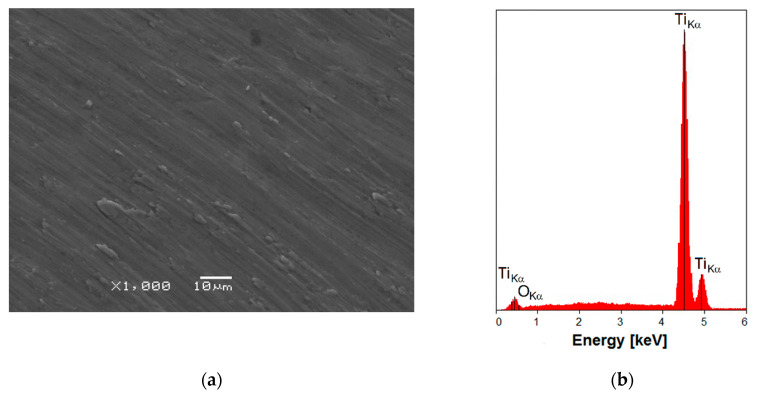
(**a**) SEM image of the Ti G4 machine surface; (**b**) EDS spectrum in the micro-region of the machine surface.

**Figure 3 materials-13-04154-f003:**
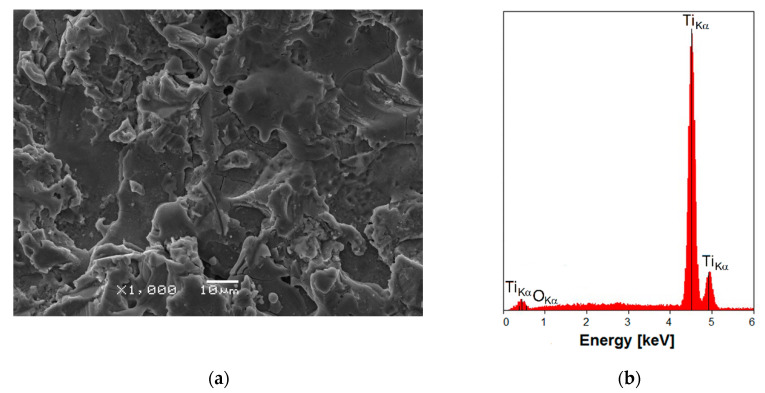
(**a**) SEM image of the Ti G4 machine surface after sandblasting; (**b**) EDS spectrum in the micro-region of the sandblasted surface.

**Figure 4 materials-13-04154-f004:**
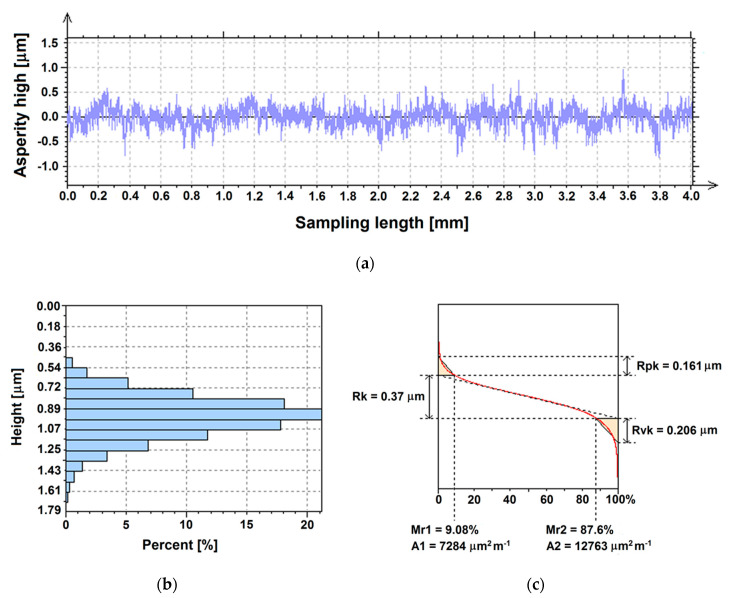
Sample profile recorded for the Ti G4 with machine surface: (**a**) Roughness profile; (**b**) Distribution of the height ordinate; (**c**) Surface bearing area ratio curve.

**Figure 5 materials-13-04154-f005:**
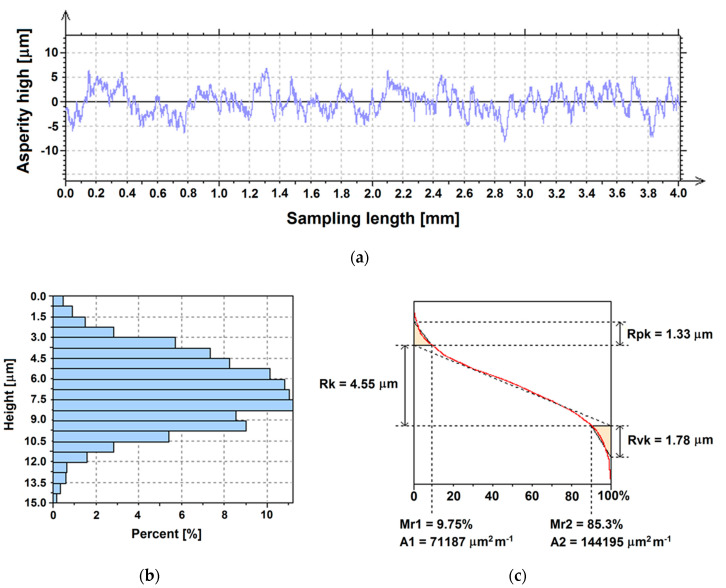
Sample profile recorded for the Ti G4 after sandblasting: (**a**) Roughness profile; (**b**) Distribution of the height ordinate; (**c**) Surface bearing area ratio curve.

**Figure 6 materials-13-04154-f006:**
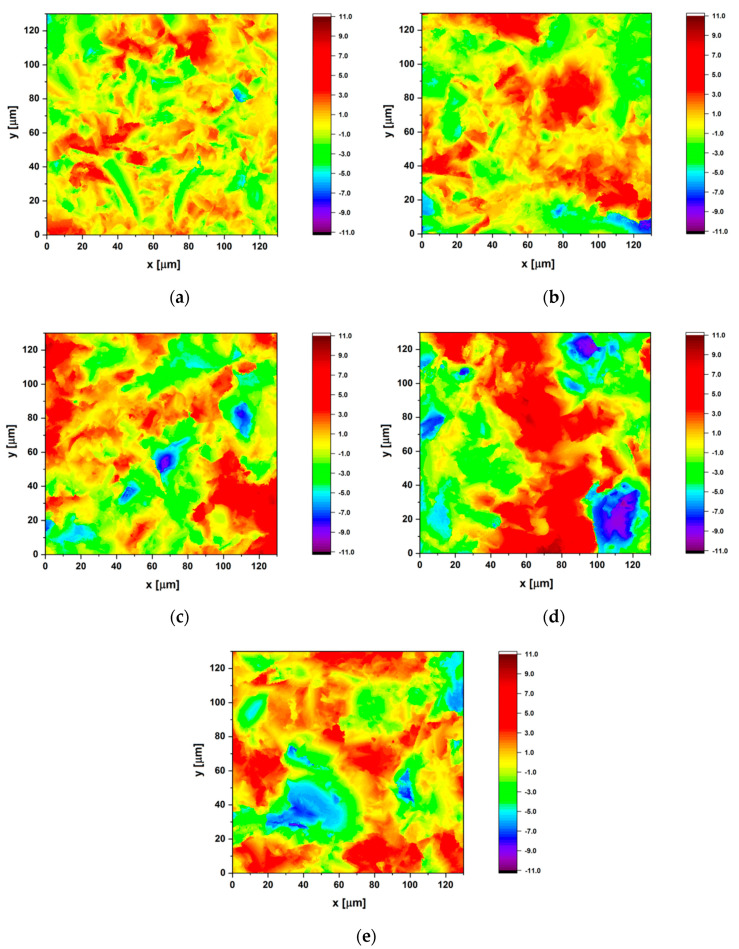
Surface topography map (130 × 130 µm^2^) obtained for the Ti G4 subjected to: (**a**) Sandblasting; (**b**) Sandblasting and autoclaving for 30 min; (**c**) Sandblasting and autoclaving for 60 min; (**d**) Sandblasting and autoclaving for 90 min; (**e**) Sandblasting and autoclaving for 120 min.

**Figure 7 materials-13-04154-f007:**
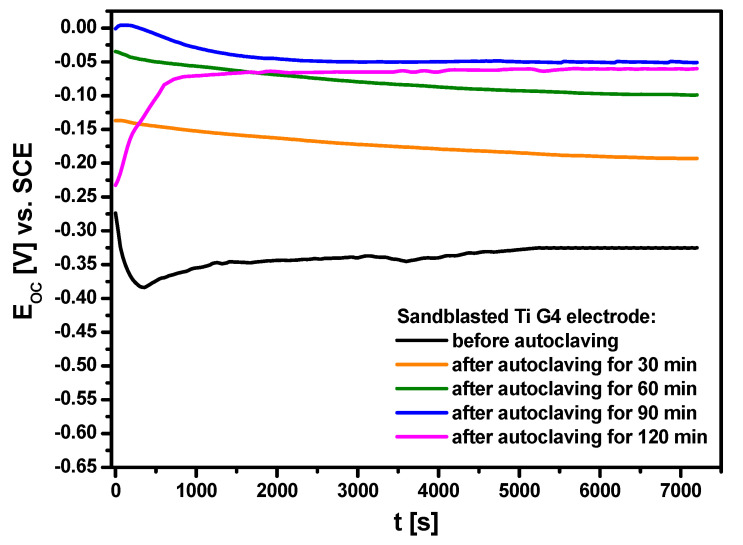
Open circuit potential as a function of immersion time for the sandblasted Ti G4 electrode in ASS at 37 °C.

**Figure 8 materials-13-04154-f008:**
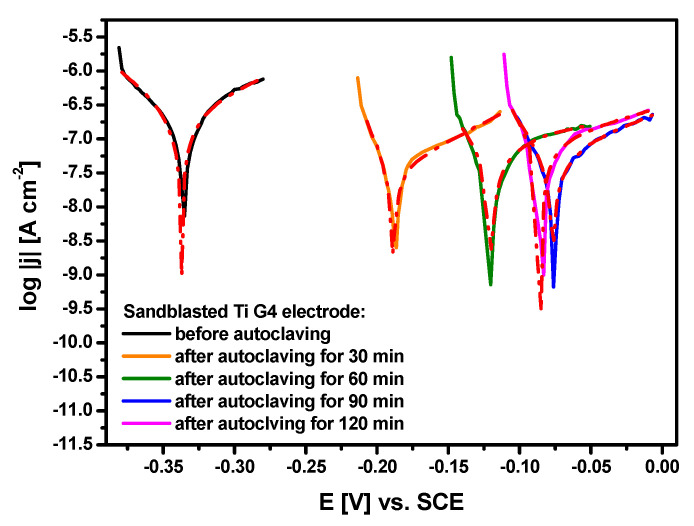
Polarization curves near the *E*_OC_ obtained for the sandblasted Ti G4 electrode before and after autoclaving in ASS at 37 °C. Experimental data are marked as continuous lines and fitted data using the Tafel extrapolation as red dash dotted lines.

**Figure 9 materials-13-04154-f009:**
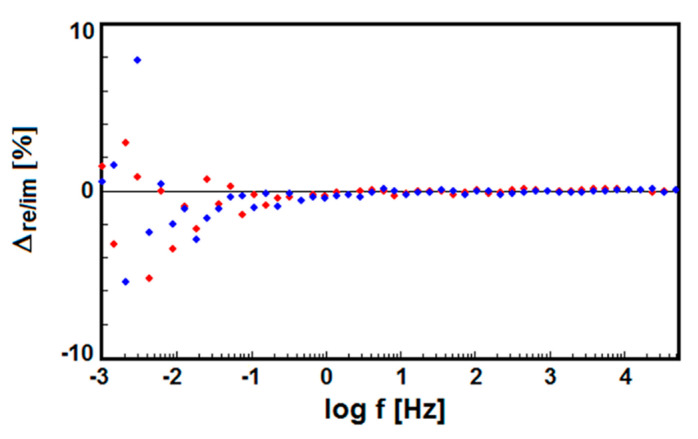
The K–K test residuals for a K–K test of the raw EIS data for the sandblasted Ti G4 electrode obtained at the *E*_cor_ in ASS at 37 °C. The red dots present the real part differences and the blue dots the imaginary part differences.

**Figure 10 materials-13-04154-f010:**
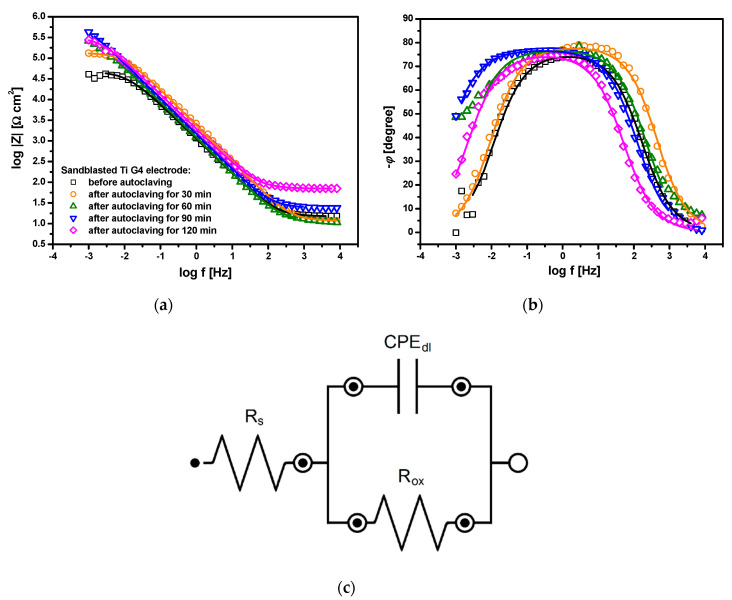
Bode diagrams obtained for the sandblasted Ti G4 electrode before and after autoclaving at *E*_cor_ in ASS at 37 °C: (**a**) Magnitude; (**b**) Phase angle; (**c**) Equivalent electrical circuit for the corrosion used to model the EIS data. Symbols are experimental data and continuous lines are CNLS fit.

**Figure 11 materials-13-04154-f011:**
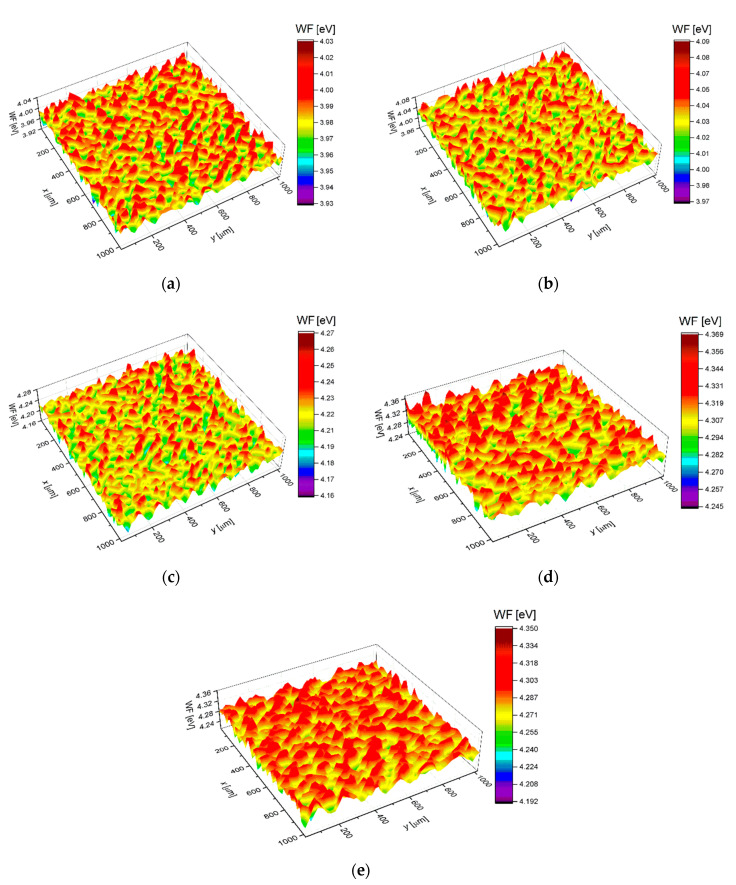
Surface distribution map of the work function for the sandblasted Ti G4: (**a**) Before autoclaving; (**b**) After autoclaving for 30 min; (**c**) After autoclaving for 60 min; (**d**) After autoclaving for 90 min; (**e**) After autoclaving for 120 min.

**Figure 12 materials-13-04154-f012:**
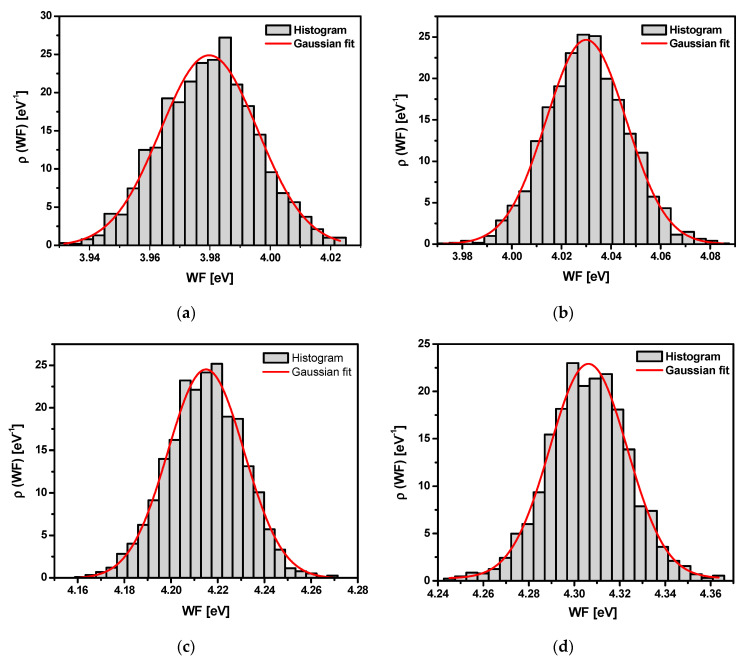
Work function distribution histogram and Gaussian fitting curve corresponding to Figure 11 for the sandblasted Ti G4: (**a**) Before autoclaving; (**b**) After autoclaving for 30 min; (**c**) After autoclaving for 60 min; (**d**) After autoclaving for 90 min; (**e**) After autoclaving for 120 min.

**Table 1 materials-13-04154-t001:** Chemical composition of Ti G4.

C	Fe	O	H	N	Ti
<0.080%	<0.500%	<0.400%	<0.008%	<0.050%	98.962%

**Table 2 materials-13-04154-t002:** Mechanical properties of Ti G4.

Tensile Strength (MPa)	Yield Point (MPa) 0.2%	HV5 Hardness	Elongation
>800	>700	>280	>10%

**Table 3 materials-13-04154-t003:** Physical properties of Ti G4.

Melting Temperature	Density	Modulus of Elasticity
1610 °C	4.5 g cm^−3^	114 GPa

**Table 4 materials-13-04154-t004:** Chemical composition of white corundum used for the sandblasting of Ti G4.

Component	Content (%)
Al_2_O_3_	99.702
SiO_2_	0.035
Fe_2_O_3_	0.045
TiO_2_	0.008
CaO	0.02
Na_2_O	0.19

**Table 5 materials-13-04154-t005:** Chemical composition of artificial saliva solution.

Component	Content (g dm^−3^)
NaCl	0.70
KCl	1.20
Na_2_HPO_4_	0.26
NaHCO_3_	1.50
KSCN	0.33

**Table 6 materials-13-04154-t006:** Basic surface texture parameters for the Ti G4 with machine and sandblasted surface according to ISO 4287 [59], where: *Pa*–arithmetic mean deviation of the primary profile, *Pq*–root mean square deviation of the primary profile, *Pp*–maximum peak height of the primary profile, *Pv*–maximum valley depth of the primary profile, *Pt*–total height of the primary profile, *Psk*–skewness of the primary profile, *Pku*–kurtosis of the primary profile, *Ra*-arithmetic mean deviation of the roughness profile, *Rz*-maximum height of the roughness profile, and *Rp*-maximum peak height of the roughness profile.

Parameter	Ti G4			
	Machine Surface		Sandblasted Surface	
	Value	Standard Deviation	Value	Standard Deviation
***Pa*** **(µm^2^ m^−1^)**	0.13	0.01	1.89	0.11
***Pq*** **(µm)**	0.17	0.01	2.33	0.12
***Pp*** **(µm]**	0.87	0.28	7.23	1.02
***Pv*** **(µm)**	0.78	0.03	7.55	0.96
***Pt*** **(µm]**	1.65	0.25	14.80	1.94
***Psk***	−0.07	0.35	−0.08	0.10
***Pku***	4.34	1.35	2.79	0.39
***Ra*** **(µm)**	0.12	0.01	1.65	0.07
***Rz*** **(µm]**	1.23	0.11	11.20	0.83
***Rp*** **(µm)**	0.69	0.17	5.37	0.51

**Table 7 materials-13-04154-t007:** *Sa* parameter defined as arithmetical mean height of the scale limited surface evaluated over the complete 3D surface in accordance with ISO 25178–2 [63] for the Ti G4 before and after autoclaving.

Type of Sample	*Sa* (μm)	SD
Ti G4 sandblasted	1.34	0.07
Ti G4 sandblasted and autoclaved for 30 min	1.84	0.09
Ti G4 sandblasted and autoclaved for 60 min	2.16	0.11
Ti G4 sandblasted and autoclaved for 90 min	3.14	0.16
Ti G4 sandblasted and autoclaved for 120 min	2.21	0.11

**Table 8 materials-13-04154-t008:** Corrosion resistance parameters determined based on the Tafel extrapolation method of the polarization curves for the Ti G4 electrodes before and after autoclaving immersed in ASS at 37 °C (see Figure 8).

Sandblasted Ti G4	*E*_cor_ (V)	*j*_cor_ (A cm^−2^)	*b*_c_ (V dec^−1^)	*b*_a_ (V dec^−1^)	*R*_p_ (Ω cm^2^)	*CR* at *E*_cor_ (mm yr^−1^)
Before autoclaving	−0.337(5)	2.88(30) × 10^−7^	0.073(5)	0.124(12)	6.92 × 10^4^	2.51 × 10^−3^
After autoclaving for 30 min	−0.189(5)	3.55(3) × 10^−8^	0.025(2)	0.093(5)	2.39 × 10^5^	3.09 × 10^−4^
After autoclaving for 60 min	−0.120(4)	1.01(12) × 10^−7^	0.058(6)	0.324(77)	2.10 × 10^5^	8.77 × 10^−4^
After autoclaving for 90 min	−0.077(4)	5.67(3) × 10^−8^	0.040(2)	0.117(5)	2.30 × 10^5^	4.93 × 10^−4^
After autoclaving for 120 min	−0.085(5)	8.57(5) × 10^−8^	0.034(2)	0.158(10)	1.42 × 10^5^	7.46 × 10^−4^

**Table 9 materials-13-04154-t009:** The value of parameters with their standard deviations obtained using the equivalent electrical circuit model for corrosion shown in Figure 11c to approximate the experimental EIS data for the sandblasted Ti G4 electrode before and after autoclaving in ASS at 37 °C.

Sandblasted Ti G4	*R*s (Ω·cm^2^)	*T*_dl_ (F cm^−2^ s *^ϕ^*^−1^)	*ϕ_d_* _l_	*R*_ox_ (Ω cm^2^)	${\bar{C}}_{\mathrm{dl}}$ (F cm^−2^)	*d*_ox_ (nm)
Before autoclaving	14.90(11)	1.88(1) × 10^−4^	0.844(2)	4.81(7) × 10^4^	6.30 × 10^−5^	1.2
After autoclaving for 30 min	11.64(14)	8.49(7) × 10^−5^	0.872(2)	1.34(2) × 10^5^	3.08 × 10^−5^	1.3
After autoclaving for 60 min	12.03(37)	1.71(2) × 10^−4^	0.858(3)	2.30(23) × 10^5^	6.13 × 10^−5^	1.4
After autoclaving for 90 min	24.24(13)	1.41(1) × 10^−4^	0.856(1)	8.80(28) × 10^5^	5.43 × 10^−5^	2.0
After autoclaving for 120 min	70.20(67)	1.11(1) × 10^−4^	0.840(2)	3.36(8) × 10^5^	4.39 × 10^−5^	3.0

**Table 10 materials-13-04154-t010:** Gaussian fitting parameters of the work function according to Equation (11) for the sandblasted Ti G4 before and after autoclaving.

Sandblasted Ti G4	$\bar{x}$	σ^2^
Before autoclaving	3.98	(0.02)^2^
After autoclaving for 30 min	4.03	(0.02)^2^
After autoclaving for 60 min	4.22	(0.02)^2^
After autoclaving for 90 min	4.31	(0.02)^2^
After autoclaving for 120 min	4.28	(0.02)^2^

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
