# Peer review of "Effect of Autoclaving Time on Corrosion Resistance of Sandblasted Ti G4 in Artificial Saliva"

_materials, 2020, doi:10.3390/ma13184154_

Round 1

Reviewer 1 Report

This paper discussing the corroison resistance enhancement by sandblasting with Al2O3 is generally well organized and clearly presented. Therefore, the reviewer would like to suggest the publication of the work after minor revision.

1, the background in the introduction could be updated with more recent papers, to further clarify the current progress and also make the innovation of this work clearer, such as [Clin. Oral Impl. Res. 29, 2018; Surf. Coatings. Tech. 2019, 180-188], etc.

2, The thickness of Al2O3 should be further discussed and well controlled, since this would be a vital parameter in this method;

3, Regarding to the roughness of the coating, although it provides the good property for resisting corrosion, would it be easy to attach bacterial? Could the authors provide some proof?

Author Response

Reviewer’s comments:

We are very grateful to the Reviewer for detailed comments on our submission. We have answered all the Reviewer’s remarks and corrected the manuscript accordingly. The detailed rebuttal is shown below.

This paper discussing the corroison resistance enhancement by sandblasting with Al2O3 is generally well organized and clearly presented. Therefore, the reviewer would like to suggest the publication of the work after minor revision.

1, The background in the introduction could be updated with more recent papers, to further clarify the current progress and also make the innovation of this work clearer, such as [Clin. Oral Impl. Res. 29, 2018; Surf. Coatings. Tech. 2019, 180-188], etc.

As requested by the Reviewer, the background in the Introduction was updated with very recent papers as following:

Ref 1. Nicholson, J.W. Titanium alloys for dental implants: A review. Prosthesis 2020, 2, 100-116, DOI: 10.3390/prosthesis2020011.

Ref 3. Lobat, T. Applications of Biomedical Engineering in Dentistry; Publisher: Springer Nature, Switzerland, 2020; ISBN 978-3-030-21582-8, DOI: 10.1007/978-3-030-21583-5.

Ref 4. Verma, R.P. Titanium based biomaterial for bone implants: A mini review. Mater. Today - Proc., 2020, DOI: 10.1016/j.matpr.2020.02.649.

Ref 8. Kapoor, N.; Nagpal, A.; Verma, R. Surface Treatment of Titanium Implant and Dental Implant Design: Titanium as biomaterial and method used for surface treatment to increase bone and soft tissue integration; Publisher: LAP LAMBERT Academic Publishing, Mauritius, 2020; ISBN 978-620-2-55638-5.

Ref 10. Singh, H.; Singh, S.; Prakash, C. Current trends in biomaterials and bio-manufacturing. Biomanufacturing 2019, 1-34, DOI: 10.1007/978-3-030-13951-3_1. 

2, The thickness of Al2O3 should be further discussed and well controlled, since this would be a vital parameter in this method;

The authors agree with the Reviewer's opinion. The type and particle size of the abrasive used for sandblasting play an important role in the shaping of the titanium surface [23-26]. The most commonly used material for sandblasting titanium dental implants is Al2O3 with the grain size ranging from 25 to 250 μm. Increasing the grain size results in an increase in surface roughness. However, it has been proved that the best results of osseointegration are obtained with a grain size of 25-75 µm. The surface obtained in this way is rough enough to ensure the desired interlocking with the bone tissue, but not too rough to release ions. In the presented study, the authors used white corundum of FEPA Grit F220 with the average micron size of the grain 53 mm.

3, Regarding to the roughness of the coating, although it provides the good property for resisting corrosion, would it be easy to attach bacterial? Could the authors provide some proof?

The authors thank the Reviewer for this valuable comment. We expect an influence of the surface roughness on bacterial adhesion. However, the antibacterial and biological properties of sandblasted and autoclaved titanium will be characterized by us in the next stage of the research. The literature describes research on bacterial adhesion to sandblasted titanium [Wang, X.; Wang, G.; Liang, J.; Cheng, J.; Ma, W.; Zhao, Y. Staphylococcus aureus adhesion to different implant surface coatings: An in vitro study. Surf. Coat. Technol. 2009, 203, 3454-3458, DOI: 10.1016/j.surfcoat.2009.05.009]. To evaluate the susceptibility of the sandblasted titanium surface to bacterial adhesion, Staphylococcus aureus (S. aureus), a major pathogen often found in the percutaneous implantassociated infections, was used as the model for the in vitro study. It was found that S. aureus adherence to the sandblasted Ti was higher than that to the un-modified surface of titanium. This results suggested promotion of S. aureus adhesion and possibility of higher occurrence of infection in vivo after sandblasting. To the best of our knowledge, no comparable tests are available for sandblasted and autoclaved CpTi G4.

Reviewer 2 Report

This report investigates the impacts of autoclaving duration on corrosion resistance of Sandblasted Ti G4 with a series of approaches including SEM, EDS, CLSM, EIS, etc. It was concluded that sterilization in a steam autoclave is an effective method of surface modification of the sandblasted Ti G4 dental metal enhancing its corrosion resistance in the ASS with pH of 7.4 at 37 °C. This manuscript is overall well-assembled and can provide useful reference for researchers in the field. A few minor issues are listed below.

  1. Figure 6 is barely discussed. More explanations should be included. The authors may also consider unifying the scale of the subfigures so that the images can be directly compared with each other.
  2. In section 3.5, equation 4-5 and corresponding descriptions are questionable as Ti4+ and O2- are not commonly individual cation/anion. It should be clarified what species are involved.
  3. For figures 4-5, are the color, grid, border, and layout optimized? Their style seems different than other figures in the manuscript.
  4. A high volume of data from various techniques is reported in this study. The authors may consider including a designated summarizing statement describing all the key experimental results and their correlation with each other.

Author Response

Reviewer’s comments:

We are very grateful to the Reviewer for detailed comments on our submission. We have answered all the Reviewer’s remarks and corrected the manuscript accordingly. The detailed rebuttal is shown below.

Reviewer 3 Report

I have reviewed the article of materials-907244, entitled “Effect of Autoclaving Time on Corrosion Resistance 2 of Sandblasted Ti G4 in Artificial Saliva”. The manuscript was well organized and written in detail. Nevertheless, I have found that there are critical deficiencies which should be extensively supplemented to further consider the paper for publication. Several examples for those (but not limited to) can be summarized as follows.

#1. An abstract should be written concisely and state the purpose of the research, materials and methods, results and major conclusions in harmony. However, the authors have devoted lines to the backgrounds and methods, comparing to only 3 lines of the results in the abstract.

#2. The authors explain or interpret their main results and finally draw conclusions by assuming the presence, or formation of passive layers after autoclaving. The explanations and conclusions should be supported by the rigid results. As stated by the authors (line 131), the main purpose of this paper was to assess the effect of autoclaving time on the corrosion resistance of sandblasted titanium. Therefore, the properties of the oxide films such as chemical composition and thickness after autoclaving should be characterized using X-ray photoelectron spectroscopy and/or transmission electron microscopy, etc. The oxide layers will be formed during the autoclaving and the properties of the oxides will be affected by the autoclaving conditions, resulting in a change in corrosion resistance of the test material.

#2. As stated by the authors (line 131), the main purpose of this paper was to assess the effect of autoclaving time on the corrosion resistance of sandblasted titanium. Oxide layers will be formed during the autoclaving and the properties of the oxides will be affected by the autoclaving conditions, resulting in a change in corrosion resistance of the test material. However, the authors explain or interpret their main results and finally draw conclusions by assuming the presence, or formation of passive layers after autoclaving, without providing any data for the oxide layers formed after the autoclaving. The properties of oxides formed on a metal surface are directly correlated with the corrosion resistance of the metal. Therefore, the properties of the oxide films such as chemical composition and thickness after autoclaving should be characterized using X-ray photoelectron spectroscopy and/or transmission electron microscopy, etc. The explanations and conclusions should be supported by the rigid results.  

#3. Based on the polarization curves in Figure 9, the following statement is not correct (line 547): “When the Eb potential is exceeded, the current density decreases rapidly and then becomes constant despite the further increase in the anode potential, which corresponds to a transpassive behavior.” According to the authors, the Eb is the electrochemical potential at which the beakdown of the oxide occurs and is used as a measure of susceptibility to pitting corrosion. By the definition of the Eb, if the potential exceeds the Eb, the current density should increase abruptly and clearly due to the breakdown of the protective oxide layers and the resultant intense anodic dissolution of the anode material. However, no one can observe the phenomena in the curves. Therefore, the statement, “When the Eb potential is exceeded, the current density decreases rapidly” is nonsense. The authors also say that the potential regions beyond about 2 VSCE correspond to a passive state. Please see the current densities in the potential regions, which are approximately 1 mA/cm2, very high values. We cannot say that a metal is in a passive state if the metal is corroding with such a high current density.

#4. The authors should show evidence of pitting corrosion by providing SEM micrographs on the surfaces of the samples after polarization scans as shown in Figure 9.

#5. There are a lot of sterilization methods. Autoclaving in steam is just one of them. The title of the manuscript is also as follows: “Effect of autoclaving time on …” However, the authors use the word “sterilization or sterilized” in the text, figures, and tables, which is inappropriate.

Author Response

(The authors gave the same response as above.)

Round 2

Reviewer 3 Report

I have reviewed the revised article of materials-907244, entitled “Effect of Autoclaving Time on Corrosion Resistance 2 of Sandblasted Ti G4 in Artificial Saliva”. The manuscript was well organized and written in detail. Nevertheless, I have found that there are still critical deficiencies which should be corrected to further consider the paper for publication.

#1. Based on the polarization curves in Figure 9, the following statement is not correct (line 547): “When the Eb potential is exceeded, the current density decreases rapidly….” According to the authors, the Eb is the electrochemical potential at which the breakdown of the oxide occurs. By the definition of the Eb, if the potential exceeds the Eb, the current density should increase abruptly due to the breakdown of the protective oxide layers and the resultant intense anodic dissolution of the anode material. However, I cannot observe the rapid current increase at the Eb in the curves. Instead, we see only the current decrease after the so-called Eb. The authors say in their response to the reviewer’s comment #4 that “The stability of TiO2 layers is lost at Eb, where the transpassivity starts. This phenomenon is manifested by an increase in anodic current densities after exceeding the Eb potential….” Therefore, the statement, “When the Eb potential is exceeded, the current density decreases rapidly” is nonsense.

#2. The authors say (line 566) that “The effect of autoclaving on the susceptibility of the sandblasted Ti G4 to pitting corrosion in the ASS at 37 ℃ was determined on the basis of anodic polarization curves (Figure 9).” They also say (line 589) that “ The obtained results indicate a lower susceptibility to pitting corrosion of the autoclaved electrodes compared to the sandblasted Ti G4 before autoclaving.” The reviewer’s comment #4 was like this: The authors should show evidence of pitting corrosion by providing SEM micrographs on the surfaces of the samples after polarization scans as shown in Figure 9. However, the authors responded to the comment as follows: “The anodic polarization curves show in Figure 9 were recorded from a potential 150 mV more negative relative to the EOC to 4 V vs. SCE. It follows that the electrochemical measurements were completed when the tested electrodes were in the transpassive range. There was no break-down of the transpassive layer, so the observation of pitting with SEM was impossible.” It is obvious that they contradict themselves. Therefore the authors should elaborate the statements line 566 and 589. How do you evaluate the pitting corrosion resistance in Figure 9 where pitting potential was not observed. (The polarization curves were obtained within the transpassive range)

Author Response

We are very grateful to the Reviewer for detailed comments on our submission. We have answered all the Reviewer’s remarks and corrected the manuscript accordingly. The detailed rebuttal is shown below.

Round 3

Reviewer 3 Report

My second comment in the 2nd peer review process was as follows. 

[The authors say (line 566) that “The effect of autoclaving on the susceptibility of the sandblasted TiG4 to pitting corrosion in the ASS at 37 ℃ was determined on the basis of anodic polarization curves (Figure 9).” They also say (line 589) that “ The obtained results indicate a lower susceptibility to pitting corrosion of the autoclaved electrodes compared to the sandblasted Ti G4 before autoclaving.”
The reviewer’s comment #4 was like this: The authors should show evidence of pitting corrosion by providing SEM micrographs on the surfaces of the samples after polarization scans as shown in Figure 9. However, the authors responded to the comment as follows: “The anodic polarization curves show in Figure 9 were recorded from a potential 150 mV more negative relative to the EOC to 4 V vs. SCE. It follows that the electrochemical measurements were completed when the tested electrodes were in the transpassive range. There was no break-down of the transpassive layer, so the observation of pitting with SEM was impossible.” It is obvious that they contradict themselves. Therefore the authors should elaborate the statements line 566 and 589. How do you evaluate the pitting corrosion resistance in Figure 9 where pitting potential was not observed. (The polarization curves were obtained within the transpassive range).]

However, the authors responded to the comment as follows (the same answer as before).

[The anodic polarization curves shown in Figure 9 were recorded from a potential 150 mV more negative relative to the EOC to 4 V vs. SCE. It follows that the electrochemical measurements were completed when the tested electrodes were in the transpassive range. There was no break-down of the transpassive layers. The observation of pitting with SEM was impossible, which is consistent with the results presented in Figure 9.]

In summary,  the authors evaluate the pitting susceptibility from Figure 9 (line 589) but they also say that "There was no break-down of the transpassive layers. The observation of pitting with SEM was impossible, which is consistent with the results presented in Figure 9. (above reponse)"

It is obvious that they still contradict themselves.

Author Response

(The authors gave the same response as above.)
